# Spatial–Temporal Variation of Cropping Patterns in Relation to Climate Change in Neolithic China

**Ruo Li [1], Feiya Lv [1,2], Liu Yang [1], Fengwen Liu [1,3], Ruiliang Liu [4,*] 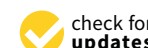 and Guanghui Dong [1,5,*]**

1   Key Laboratory of Western China's Environmental System (Ministry of Education), College of Earth and Environmental Sciences, Lanzhou University, Lanzhou 730000, China; lir2014@lzu.edu.cn (R.L.); lvfy14@lzu.edu.cn (F.L.); lyang2018@lzu.edu.cn (L.Y.); liufw@ynu.edu.cn (F.L.)
2   Department of Geography, University of Wisconsin, Madison, WI 53706, USA
3   School of Resource Environment and Earth Science, Yunnan University, Kunming 650000, China
4   School of Archaeology, University of Oxford, Oxford OX1 3TG, UK
5   CAS Center for Excellence in Tibetan Plateau Earth Sciences, Chinese Academy of Sciences (CAS), Beijing 100101, China
*   Correspondence: ruiliang.liu@arch.ox.ac.uk (R.L.); ghdong@lzu.edu.cn (G.D.)

**Abstract:** The Neolithic period witnessed the start and spread of agriculture across Eurasia, as well as the beginning of important climate changes which would take place over millennia. Nevertheless, it remains rather unclear in what ways local societies chose to respond to these considerable changes in both the shorter and longer term. Crops such as rice and millet were domesticated in the Yangtze River and the Yellow River valleys in China during the early Holocene. Paleoclimate studies suggest that the pattern of precipitation in these two areas was distinctly different. This paper reviews updated archaeobotanical evidence from Neolithic sites in China. Comparing these results to the regional high-resolution paleoclimate records enables us to better understand the development of rice and millet and its relation to climate change. This comparison shows that rice was mainly cultivated in the Yangtze River valley and its southern margin, whereas millet cultivation occurred in the northern area of China during 9000–7000 BP. Both millet and rice-based agriculture became intensified and expanded during 7000–5000 BP. In the following period of 5000–4000 BP, rice agriculture continued to expand within the Yangtze River valley and millet cultivation moved gradually westwards. Meanwhile, mixed agriculture based on both millet and rice developed along the boundary between north and south. From 9000–7000 BP, China maintained hunting activities. Subsequently, from 7000–6000 BP, changes in vegetation and landscape triggered by climate change played an essential role in the development of agriculture. Precipitation became an important factor in forming the distinct regional patterns of Chinese agriculture in 6000–4000 BP.

**Keywords:** Yangtze River valley; Yellow River valley; rice cultivation; millet cultivation; precipitation; Neolithic China

## 1. Introduction

The climate is one of the driving forces behind the social evolution of humans, especially in prehistory [1–4]. The studies of different strategies adopted by human societies in response to drastic climate fluctuation in the past can provide valuable insights into the underlying patterns and mechanisms of the human–land relationship. They can also offer important lessons on coping with the current challenges of rapid climate change in the modern world, such as global warming. The Neolithic period coincided with the early-mid Holocene—a recent warming period with numerous considerable climate fluctuations. It is one of the most significant stages of sociocultural evolution in human history.

In recent years, there has been an increase of research focusing on human–environment interaction in this specific period [4–8].

One of the era-defining events in the Neolithic was the development of agriculture across the old world, followed by a substantial increase in the size of population and settlements [9–12]. Climate change has been considered to be a critical factor in the emergence and intensification of agriculture during the Neolithic period [13,14]. While the changes in temperature follow the same approximate trends in different regions of the Northern Hemisphere [15], precipitation shows distinct patterns affected by local climate (e.g., the arid central Asia and the Asian Monsoon Region) [16,17].

East Asia was one of the points of origin of agricultural development. Rice and broomcorn/foxtail millet were domesticated around 10,000 BP in the Yangtze River valley (southern China) and the Yellow River valley (northern China), respectively [18–20]. Both crops later became widely cultivated and formed the well-known agricultural structure in Neolithic China of northern millet vs southern rice [21]. Paleoclimate studies also suggest a similar geographical distinction between the Yangtze and the Yellow River valley in terms of moisture variation, called the anti-phase pattern [22]. The relationship between agricultural development in China and the regional variation of precipitation has not yet been discussed in detail. In this paper, we have collected and analyzed a large amount of legacy data, containing both archaeobotanical remains and radiocarbon dates from 125 Neolithic sites in China. Correlating the archaeobotanical evidence with the variation of precipitation reconstructed from well-dated paleoclimate records fills a significant gap in the current literature and allows us re-explore the different phases of Neolithic China in more detail.

## 2. Spatial–Temporal Change of Human Cropping Structures in Neolithic China

It has been widely agreed that the Neolithic cultural evolution in northern China can be divided into three phases: the pre-Yangshao period (9000–7000 BP), the Yangshao period (7000–5000 BP) and the Longshan period (5000–4000 BP). This chronological framework is based on multiple lines of evidence ranging from material culture (e.g., pottery typology), settlement, dietary practice, technological change, stratigraphy and radiocarbon dating [23–25]. It offers a crucial point of departure for us to reconstruct spatial patterns of the development of agriculture in China based on the archaeobotanical evidence, which in return contributes to the overall picture. We collected archaeobotanical data from 125 Neolithic sites across China: the data recorded include the presence and absence of targeted crops (millet, rice, barley, wheat), their absolute quantities and ratios between different types of crops. Soybean crops have been excluded from this work, although carbonized grains have occasionally been identified at some Neolithic sites in China, owing to the difficulty in separating domesticated from wild varieties.

As is the case with legacy collections, the quality of data varies considerably. The first level of complexity is caused by the scale of the archaeological excavation, as a fully excavated site might present a more complete picture of crops in use at a given time in comparison to samples collected from a surface survey. The data may have also derived from different theoretical and methodological approaches prioritizing certain practices or scrutiny above others, and finally the identification of crop remains is often subject to the individual analyst's experience (e.g., wild crops vs domesticated crops). Despite these potential complexities, the broad changes indicated by our big-data approach are arguably valid and important, as the broad patterns are not based on any specific site or site type and there are always multiple sites within one region for cross-checking. Furthermore, additional data, such as stable isotopes, are also included in the discussion and help to counter-check the archaeological narratives.

## 3. Spatial Pattern of Cropping Structures in China during 9000–7000 BP

Archaeobotanical studies suggest that the initial domestication of rice took place in the middle reaches of the Yangtze River prior to 10,000 BP, whereas broomcorn and foxtail millet were first domesticated in the Central Plains of northern China around 10,000 BP [18–20,26]. However, due to the limited number of identifiable crop fossils in these early Neolithic sites and the lack of direct radiocarbon

dating, the exact chronology of the earliest millet/rice remains highly controversial. As macrofossils of domesticated millet and rice become more ubiquitous in sites dated between 9000 and 7000 BP and have direct radiocarbon dates associated with them, it is possible to say that the cultivation of rice and millet dated at least as far back as 9000 BP [19,27–30]. Macro and micro fossils of rice dated to 9000–7000 BP were unearthed from sites in the middle and lower reaches of the Yangtze River, such as Shangshan, Xiaohuangshan and Pengtoushan [20]. These sites are mainly in the piedmont zones, possibly owing to locational convenience for hunting and gathering activities, which were the primary survival strategies during this period. Interestingly, rice appears also to be the most ubiquitous crop in a few sites in the southern margin of the middle Yellow River valley (Figure 1), as exemplified by the famous Jiahu site, one of the largest settlements of the pre-Yangshao culture in the present-day Henan Province [31,32].

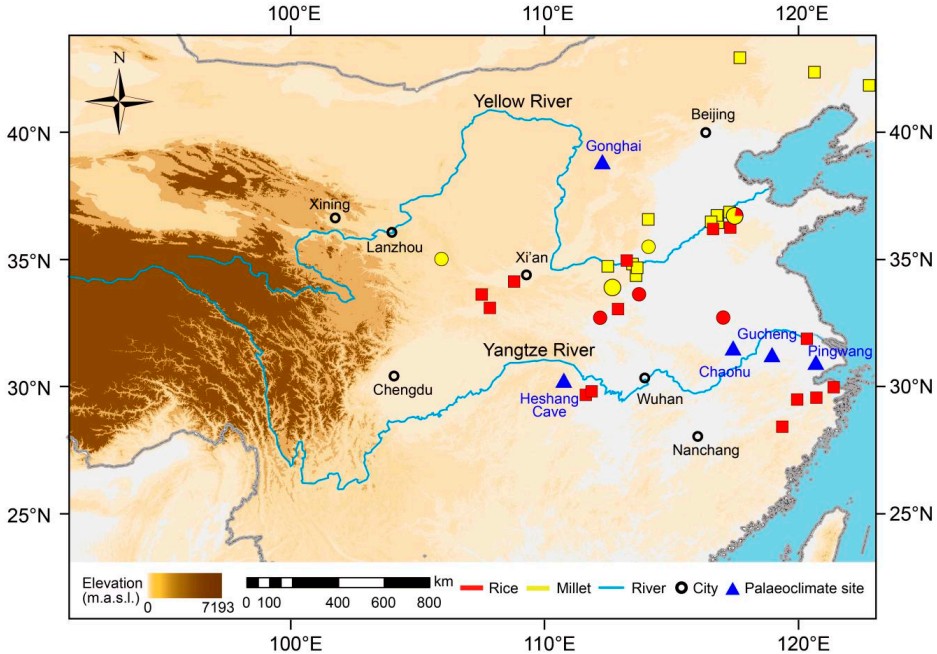

**Figure 1.** The spatial pattern of crop macro-fossils from sites dated between 9000–7000 BP in China. Squares represent sites without detailed archaeobotanical data; circles represent sites with detailed archaeobotanical data.

Charred grains of broomcorn and foxtail millet have been identified from numerous pre-Yangshao cultures (9000–7000 BP) in northern China, including the Peiligang, Xinglongwa, Houli, Cishan and Laoguantai-Dadiwan [19,27,28,33]. In some sites of the Peiligang and Houli cultures, both millet and rice were cultivated together (Figure 1). Most pre-Yangshao sites in northern China were located in the foothill areas, probably to facilitate hunting and gathering [34]. Stable carbon isotopes of human bones show a clear $C_3$ signal [35,36], indicating that hunting and gathering—rather than millet cultivation—was the major food strategy in this period [37]. Based on the large number of charred grains of foxtail and broomcorn millet as well as the clear $C_4$ signal from the isotope analysis of human bones in the sites of Xinglonggou and Xinglongwa in east Inner Mongolia [38], it is possible that millet cultivation might have become the principal food strategy in northern China during 8000–7000 BP [39].

## 4. Spatial Pattern of Human Cropping Structures in China during 7000–5000 BP

Evidence from the Yangshao Period sites in China (7000–5000 BP) suggests that, by this time, crop cultivation had replaced hunting/gathering and was the primary subsistence strategy in both the Yangtze and Yellow River valleys; for example, huge amounts of rice remains have been found in the storage pits of the Hemudu site [40]. Moreover, the ratio of domesticated rice to wild rice increased

rapidly between ~6700–6300 BP in the lower Yangtze River valley [41]. In the Yellow River valley, carbon isotopes of human bones unearthed from almost all Yangshao sites display a clear C$_4$ signal. This strongly indicates that millet became a routine part of diet in northern China between 7000 and 5000 BP [39].

The geographical distribution of both millet and rice from 7000 to 5000 BP is undoubtedly larger than it was from 9000 to 7000 BP (Figures 1 and 2). During the Yangshao period (7000–5000 BP), the regular practice of rice cultivation extended further northwestward to the west Loess Plateau; one example of this is the presence of domesticated rice at Xishanping, a site in the western Loess Plateau [42]. The distribution of millet expanded in several directions: eastward to the Shandong Peninsula and even Korea [43], westward to the northeast Tibetan Plateau [44], southwestward to the Chengdu Plain [45] and southward to the middle Yangtze River valley (Figure 2; [46]).

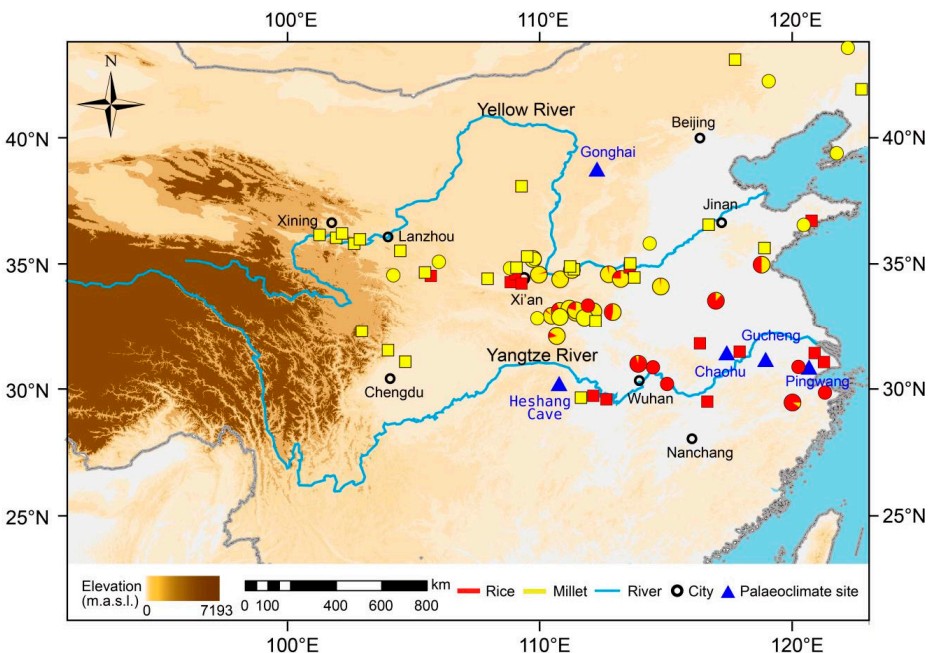

**Figure 2.** The spatial pattern of crop macro-fossils from sites dated between 7000–5000 BP in China. Squares represent sites without detailed archaeobotanical data; circles represent sites with detailed archaeobotanical data.

## 5. Spatial Pattern of Human Cropping Structures in China during 5000–4000 BP

After the Yangshao period, there was a broad westward movement of millet farming in the Longshan period (5000–4000 BP, Figures 2 and 3). In the east costal area of Shandong, local people replaced millet with rice during 5000–4000 BP. In contrast, farmers of the Machang culture in western China (4300–4000 BP) continued to rely heavily on millet cultivation. Together with the movement of farmers, millet moved gradually westward and was cultivated in the Hexi Corridor [47] and east Xinjiang [48]. Foxtail millet has also been identified at Karuo, a site on the southeastern Tibetan Plateau, which has been dated to ~4700–4300 BP [49]. One should bear in mind that these crops were also likely to be exchanged from adjacent farming societies rather than being locally cultivated [50].

The Yangtze River valley saw an expansion of rice cultivation during the Longshan period (5000–4000 BP) when rice was undoubtedly the dominant crop in the region. During the same period, it replaced millet and became the major subsistence crop in the northwest Chengdu basin as well (Figure 3; [45]). Meanwhile, rice cultivation spread westward to the Yunnan-Guizhou Plateau [51], as exemplified by the large number of rice and millet macrofossils from the first phase of the Baiyangcun site in northwest Yunnan [52]. It was further introduced into the Pearl River Delta of southern China during 5000–4000 BP. Charred rice grains have been discovered in both Laoyuan and Chaling sites [53].

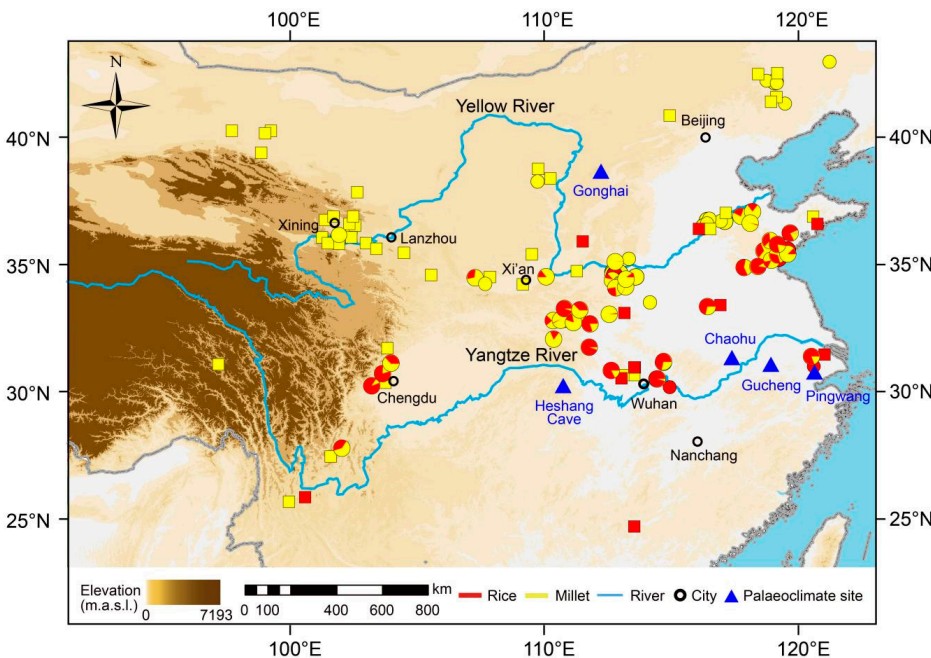

**Figure 3.** The spatial pattern of crop macro-fossils from sites dated between 5000-4000 BP in China. Squares represent sites without detailed archaeobotanical data; circles represent sites with detailed archaeobotanical data.

## 6. Spatial–Temporal Variation of Agriculture Patterns in Response to Climate Change in Neolithic China

The archaeobotanical evidence reveals a series of broad changes in the spatial patterns of agricultural development in Neolithic China. The overall area of rice cultivation appears to be larger than that of millet during 9000–7000 BP, and the boundary between these two traditional agricultural systems lay roughly east–west at ~34 °N (Figure 1). This boundary moved slightly southward to ~33 °N during 7000–5000 BP, as illustrated by the distribution of sites that yielded charred millet grains (Figure 2). During 5000–4000 BP, the boundary between rice and millet shifted to an approximate northeast–southwest direction. In east China, the northern limit for rice-based agriculture in the Shandong Peninsula moved to ~36 °N. In the Chengdu Plain of southwest China, the local cropping structure changed from a combination of broomcorn and foxtail millet to a combination of rice and foxtail millet at around 4700 BP [45]. Millet agriculture appears most dominant in the upper and middle valley of the Yellow River, the Hexi Corridor and the Yanshan-Liaoning area of northeast China (Figure 3).

Given the fact that the environmental conditions for the growth of the crops vary between species, broad changes in agriculture activities can be correlated with paleoclimate records. In order to better understand how the spatial–temporal variation of rice and millet developed, we compared the broad changes in agriculture activities with a number of key paleoclimate records in northern and central China. The record, with a ~20 year resolution precipitation reconstruction from Gonghai Lake (Figure 4f, [54]) in northern China, is indicative of gradually increasing precipitation from 14,600 to 7800 BP. Precipitation reached a maximum between 7800 and 5300 BP and decreased after 5300 BP. The generally high rainfall/moisture stage from ~8000 to ~5000 BP has also been recorded in other lake sediments and loess sections such as the Daihai Lake [55] and the Dadiwan section [56]. It is also possible to cross-check the change in precipitation with different kinds of paleoclimate records. For example, a large synthesis of precipitation based on 310 dates from 77 sites on the Loess Plateau shows that the paleosol probability density was relatively high from 8000 to 5000 BP, reflecting the relatively high moisture conditions during this period [57].

Precipitation and moisture records from the middle and lower reaches of the Yangtze river during the Holocene are relatively rare and more difficult to interpret. The pollen-based precipitation record from the Chaohu Lake [58], the Gucheng Lake [59] and the Pingwang Lake [60] in the lower Yangtze region suggest that precipitation reached its maximum between 10,000 and 7000 BP, after which it followed a broad decline with strong oscillations up to the present day (Figure 4d, [61]). Moreover, the stalagmite ARM/SIRM record from the Heshang Cave (Figure 4e, [62]) and the mass accumulation rates of hopanoids from the Dajiuhu Peat bog [62], often used to reflect the Holocene paleo-humidity variations of the middle reaches of the Yangtze River, also demonstrate that the climate became more humid between 11,000–7000 BP and 3000–1000 BP but was more arid and highly variable between 7000 and 3000 BP. Although a number of small inconsistencies concerning precipitation or moisture can be found in these records, the consensus is that there was relatively high precipitation and moisture from 10,000 to 7000 BP which decreased between 7000 and 5000 BP. Precipitation and moisture appear to have subsequently increased from 5000–4000 BP in comparison.

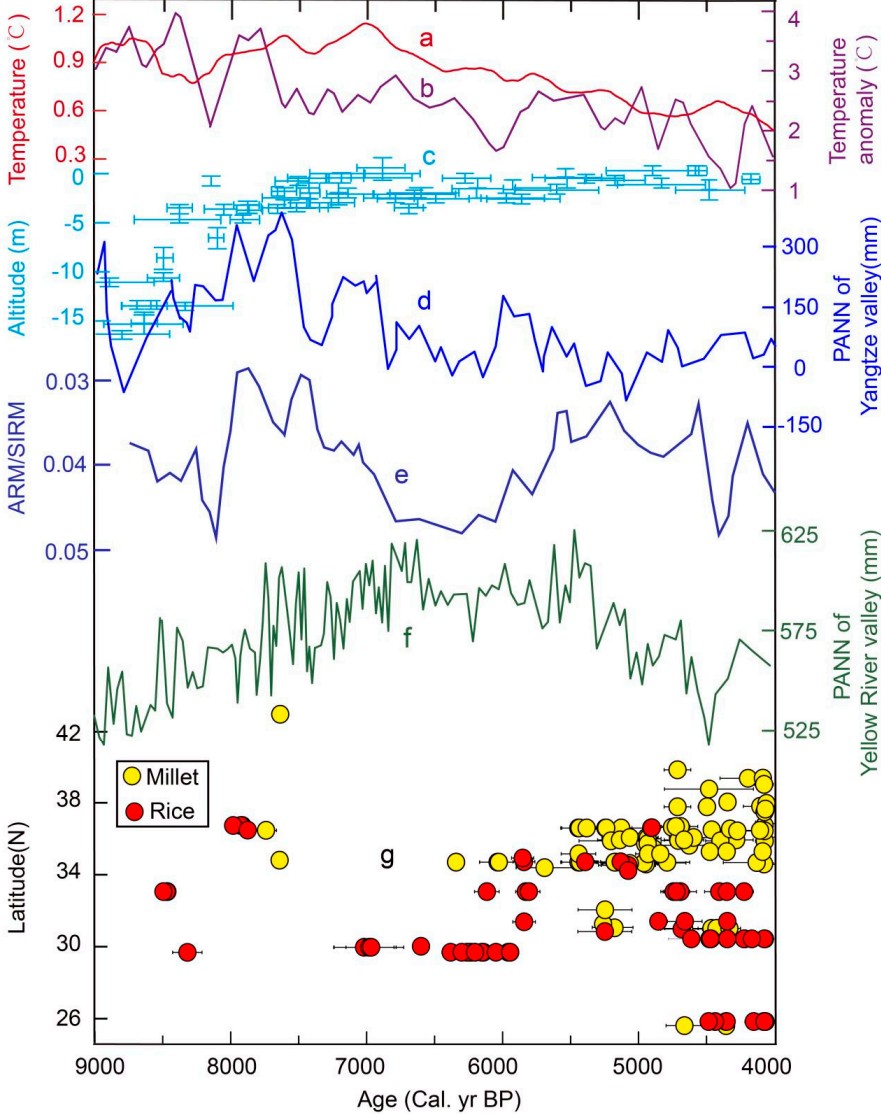

**Figure 4.** Comparison of (**a**,**b**) temperature anomaly in the Northern Hemisphere (30–90 °N) [15] and High Arctic [63]; (**c**) reconstructed sea-level change [64]; (**d**) pollen-based annual precipitation (PANN) in the Yangtze River valley [61]; (**e**) stalagmite ARM/SIRM record from the Heshang Cave [62]; (**f**) reconstructed PANN from Gonghai lake [54]; (**g**) latitudinal distribution of Neolithic sites with unearthed rice or millet sites between 9000–4000 BP.

We have considered the temperature anomaly in the Northern Hemisphere (30–90 °N) [15] and High Arctic [63] and pollen-based pollen-based annual precipitations (PANNs) for the Yangtze River region [61] and the Gonghai Lake [54], together with the stalagmite ARM/SIRM record from the Heshang Cave [62], as paleoclimate data, and have reconstructed the sea-level change [64] for comparison with the archaeobotanical results (Figure 4). Although the local paleoclimates of the region are complicated, the trend of the multiple paleoclimate records is largely valid on the regional scale. It was relatively wet in the Yangtze River valley and dry in the Yellow River valley during 9000–7000 BP and 5000–4000 BP, but these conditions changed to become completely opposite during 7000–5000 BP (Figure 4).

We suggest that the change to a relatively high temperature and precipitation during 9000–7000 BP (Figure 4a,b) provided ideal conditions for rice cultivation along the Yangtze River. Its growth requires both an appropriate temperature and adequate moisture. In comparison, millet (broomcorn and foxtail) are drought-tolerant and frost-sensitive and thus better adapted to the climate in northern China. However, the sea level transgression between 9000–7000 BP in the coastal plain of eastern China exerted considerable impact on the broad-spectrum economy based on fishing, hunting/gathering and agriculture along the lower reaches of the Yangtze River [65,66]. In the pre-Yangshao period, the expansion of millet seems to have been delayed in the Yellow River valley. The reason for this is still unclear but may be related to the low survival pressure mitigated by hunting/gathering. In this period, primitive agriculture was still in its infant stage, and hunting/gathering was the primary source of food supply in Yangtze River and Yellow River valleys [37,39].

During 7000–5000 BP, both precipitation and temperature began to decrease and the climate of the Yangtze River valley became dry and cool [15,61]. The sea level also began to lower at the same time (Figure 4c, [64,67]). The dry conditions and regression of the sea level after 7000 BP were disadvantageous for both fishing and hunting/gathering but provided an open landscape favorable to the rapid development of agriculture. During this time, the genetic characteristics of the rice remains from sites of the Hemudu culture (7000–5300 BP) tend to be stable and agricultural tools and pottery technology had also been significantly improved [68,69], both of which could also have facilitated rice cultivation and the expansion of this practice. On the other hand, powerful local societies such as Chengtoushan (c.6000 BP) and Liangzhu (5200–4300 BP) started to emerge in the middle and lower reaches of Yangtze River valley [70,71], indicating that the growth and aggregation of regional populations may have become increasingly dependent on agriculture [21]. Despite the drier and cooler climates, the changes in the sea level and human societies as a whole possibly triggered the major transition of food strategies to rice cultivation in the Yangtze River valley during 7000–5000 BP [21,66,72].

In the Yellow River valley, however, precipitation was evidently higher during 7000–5000 BP than 9000–7000 BP (Figure 4f, [54]). Thanks to the abundant water supply, millet agriculture became more intensified between 7000–6000 BP in the middle Yellow River valley [21,39]. This was also possibly related to the increasing size of the local population in northern China, which in return required more food supply and intensification in agriculture, particularly during the Yangshao period [73–75]. These human activities resulted in a sharp decline in forestation, making the area increasingly disadvantageous for hunting/gathering but favorable for the rise of millet cultivation [76]. The favorable climate might also facilitate rice cultivation in the Yellow River valley. Genetic evidence also shows a continuous movement of people from southern China to the Yellow River valey since the Yangshao period [42,77]. Meanwhile, the adoption of millet cultivation and its southward expansion, boosted by this climate change, is exemplified in the Chengdu Plain in upper Yangtze River valley between 6000–4700 BP, where millet became the major food supply [45].

During 5000–4000 BP, precipitation declined in the Yellow River valley but increased in the Yangtze River valley [54,61,62], while temperature followed a constant decline compared to 7000–5000 BP [15]. Climate deterioration in northern China appears to be one of the key reasons for the collapse of the Yangshao culture, which mainly relied on the supply of millet [54]. The relatively wet climate in the

Yangtze River valley during 5000–4000 BP [61] provided favorable conditions for rice cultivation, which might have promoted the transition of cropping patterns in the Chengdu Plain from millet agriculture to mixed rice–millet agriculture [45]. Meanwhile, mixed agriculture also began to thrive in the Huai River valley, located between the Yangtze River and the Yellow River [78]. The increasingly diversified farming activities considerably improved human adaptability to the widespread climate changes occurring between 5000 and 4000 BP [54,79].

In addition to millet and rice, during the second half of the Longshan period (ca. 4500–4000 BP), another important crop in Chinese prehistory—wheat—was introduced into the lower Yellow River valley [80]. As an exotic crop, it was initially not adopted as a major staple in Neolithic China. The introduction and cultivation of cold-tolerant barley and wheat greatly altered the cropping structures of northern China during the Bronze Age, especially in northwest China, where the altitude is much higher than in east China [44,81]. Agricultural innovation in the Tibetan Plateau featured the cultivation of barley and herding of sheep and yak, enabling local people to move into higher-elevation areas and settle in them permanently after 3600 BP when the climate changed to become cold and dry [44,82].

## 7. Conclusions

Archaeobotanical studies present the long and complicated trajectory of indigenous agricultural development in China. It is certainly not as simple (i.e., northern millet and southern rice) as noted in the historical documents but rather a dynamic process involving and responding to the key elements of climate change. The period 9000–4000 BP was characterized by the combination of rice-based agriculture in the Yangtze River valley and millet-based agriculture in the Yellow River valley, together with a series of variations in regional cropping patterns during different phases of the Neolithic Age. After 7000 BP, there was an important decline in temperature, which might have triggered the transformation of landscape and vegetation and promoted the transition from hunting/gathering to farming activities. The spatial–temporal variation of precipitation played an influential role in the shifting spatial patterns of farming activities during 6000–4000 BP, shedding more light on the issue of humans adapting to climate change in China during the Neolithic period prior to the adoption of exotic crops such as wheat and barley from the west.

**Author Contributions:** Data curation, R.L.; Investigation, R.L., L.Y. and F.L.; Methodology, F.L.; Supervision, G.D.; Visualization, L.Y.; Writing—original draft, R.L. and G.D.; Writing—review & editing, F.L. and R.L. All authors have read and agreed to the published version of the manuscript.

**Funding:** This research was funded by the National Natural Science Foundation of China (Grant Nos. 41671077 and 41825001) and supported by the 111 Project.

**Acknowledgments:** We would like to thank the three anonymous reviewers for their useful comments and Philly Howarth from the University of Oxford for editing the manuscript.

**Conflicts of Interest:** The authors declare no conflicts of interest.

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
