# Peer review of "Spatial–Temporal Variation of Cropping Patterns in Relation to Climate Change in Neolithic China"

_atmosphere, doi:10.3390/atmos11070677_

Round 1

Reviewer 1 Report

I am pleased to see that the manuscript has improved significantly since the first time I reviewed it. However, there remain several outstanding issues that I believe are necessary to address before publication. 

  1. The are numerous typographical and grammatical errors throughout the text that must be corrected before the paper is accepted. 
  2. I strongly recommend that the authors use a different map to display their archaeobotanical meta data study. Not only is the map very ugly to look at, it also makes the data difficult to read / understand. 
  3. Again, I also strongly recommend that the dataset the authors use for their study be made public and attached to this article as Supplemental Information. This is best practices in the discipline of archaeology and archaeobotany and, as this paper is the only large meta analysis of archaeobotanical remains in China, garner many citations. 

Once these issues are address, I will be happy to recommend this excellent study for publication. 

Author Response

Thanks very much for taking your time to review this manuscript. We thank the reviewers for the time and effort that they have put into reviewing the previous version of the manuscript. Their suggestions have enabled us to improve our work.

There are our detailed responses to Reviewer#1:

1.The are numerous typographical and grammatical errors throughout the text that must be corrected before the paper is accepted.

Re:Thanks for the comments. We have cleaned up the structures more thoroughly, polished the language and corrected the grammar mistakes. Dr Philly Howarth who is a native English speaker working in the Research Laboratory of Archaeology and History of Art (RLAHA), University of Oxford, has checked the wording.

2.I strongly recommend that the authors use a different map to display their archaeobotanical meta data study. Not only is the map very ugly to look at, it also makes the data difficult to read / understand.

Re:Thanks for the suggestion. In fact, we have drawn a new figure with all of the archaeobotanical sites as suggested in the response to reviewers in the last revised version, which might not be shown properly due to unknonw technical fault. We have tried the new figure suggested below. It is obviously unable able to better (or even worse) to delivery the message than the existed ones.

See the figure in the Word file below.

3.Again, I also strongly recommend that the dataset the authors use for their study be made public and attached to this article as Supplemental Information. This is best practices in the discipline of archaeology and archaeobotany and, as this paper is the only large meta analysis of archaeobotanical remains in China, garner many citations.

Re:Thanks for the useful suggestion. We have prepared the archaeobotanical data with radiocarbon dates and very glad to share our dataset through the online supplementary material.

Reviewer 2 Report

This is the second time I have reviewed this paper. They did a decent job addressing my overall concerns. The paper is written with a more general tone, and is stronger in the revised form. There are some typos and grammatical issues that still need to be addressed, but that can probably be dealt with during type-setting. 

Author Response

Thanks very much for taking your time to review this manuscript. We thank the reviewers for the time and effort that they have put into reviewing the previous version of the manuscript. Their suggestions have enabled us to improve our work.

There are our detailed responses to Reviewer#2:

Reviewer: This is the second time I have reviewed this paper. They did a decent job addressing my overall concerns. The paper is written with a more general tone, and is stronger in the revised form. There are some typos and grammatical issues that still need to be addressed, but that can probably be dealt with during type-setting.

Re:Thanks for your comments. We have polished the language again to avoid mistakes in grammar and typos.

Reviewer 3 Report

Congratulations, very important study. Word file attached.

I worked on English, but it may need some further edition after you have considered the changes I have suggested.

I made some notes in the bibliography, but it needs to be carefully checked.

Author Response

Thanks very much for taking your time to review this manuscript. We thank the reviewers for the time and effort that they have put into reviewing the previous version of the manuscript. Their suggestions have enabled us to improve our work.

There are our detailed responses to Reviewer #3:

Reviewer: Congratulations, very important study. Word file attached.I worked on English, but it may need some further edition after you have considered the changes I have suggested. I made some notes in the bibliography, but it needs to be carefully checked.

Re:Thanks for your work with English, we have reviewed the changes you suggested and made the revisions carefully.

Round 2

Reviewer 1 Report

This is the third time that I have reviewed this manuscript and I am pleased to see that many of my suggestions and comments have been incorporated into the present draft. I am pleased to see that there is now an SI attached to the main article, and the English has also been significantly improved in this version of the manuscript. My main major comment from the previous versions of the manuscript is that the basemap should be changed - the rainbow color scheme of the DEM is hard to look at and understand the data the authors are presenting. Nonetheless, the authors have not changed it in any of their revisions. If possible, I still think that the authors would benefit from redoing this figures to make them more readable, but I completely understand if that is not possible. Therefore, I recommend this manuscript be accepted after minor revisions - another reading of the text and potential improvement of the figures.

Author Response

Dear Editor and Reviewers,

Thanks very much for taking your time to review this manuscript. We thank the reviewers for the time and effort that they have put into reviewing the previous version of the manuscript. Their suggestions have enabled us to improve our work.

Reviewer :

This is the third time that I have reviewed this manuscript and I am pleased to see that many of my suggestions and comments have been incorporated into the present draft. I am pleased to see that there is now an SI attached to the main article, and the English has also been significantly improved in this version of the manuscript. My main major comment from the previous versions of the manuscript is that the base map should be changed - the rainbow color scheme of the DEM is hard to look at and understand the data the authors are presenting. Nonetheless, the authors have not changed it in any of their revisions. If possible, I still think that the authors would benefit from redoing this figures to make them more readable, but I completely understand if that is not possible. Therefore, I recommend this manuscript be accepted after minor revisions - another reading of the text and potential improvement of the figures.

Re:Thanks for the suggestion. We have redone the figures (figure 1-3), and we think it's more comfortable to change the base map, too. This article will attract more attention because of the replacement of the base map. Thanks again for your three reviews, your comments and suggestions are helpful to the improvement of the article.

This manuscript is a resubmission of an earlier submission. The following is a list of the peer review reports and author responses from that submission.

Round 1

Reviewer 1 Report

While I think the content in this paper is interesting to read, it is too underdeveloped to be published at this time. The authors attempt to review the available literature, but there are no new data presented in the study. If it is to be a review paper, then the data that are presented should be treated in a more robust way. The discussion presented is vague and lacking in detail. As a review paper, I suggest some statistical analyses of the available datasets, and a better discussion about uncertainty and reliability among the various published records. There is no discussion for how these data were handled.

The available paleoclimate records from the region are much more complicated than what the authors present. Instead, the authors only discuss very broad-scale changes. In some sections they refer to “frequent climate fluctuations”, but what are they? I’m unconvinced, based on the evidence presented, that there are any clear anti-phased precipitation patterns across the region. The authors should critically evaluate all the datasets that they present. I also do not understand the relevance of all the various records presented in Figure 4. For example, how do they interpret percentage of pollen? Any why only present pollen records as precipitation proxies? What about the stable isotope records from the region?

Many sections of the paper need to be reorganized. For example, the temperature discussion starts and stops in various sections. It should be more coherent. There are also numerous typos and grammatical mistakes, which makes the paper very difficult to read in its present form. 

Author Response

Thanks very much for taking your time to review this manuscript. We thank the reviewers for the time and effort that they have put into reviewing the previous version of the manuscript. Their suggestions have enabled us to improve our work.

Reviewer #1:

  1. The authors attempt to review the available literature, but there are no new data presented in the study. If it is to be a review paper, then the data that are presented should be treated in a more robust way. The discussion presented is vague and lacking in detail. As a review paper, I suggest some statistical analyses of the available datasets, and a better discussion about uncertainty and reliability among the various published records. There is no discussion for how these data were handled.

Re: Thank you very much for making this comment. As the reviewer pointed out, one primary objective of the paper is to offer a review of the current state of debate on the early agriculture development in China. Whilst it is true that in this paper few new data have been presented, one of the major contributions is to present broad changes in the development of agriculture in Neolithic China. Obliviously, this cannot be achieved by a handful new data from a few specific sites. Therefore, we choose a big-data approach to illustrate the overarching patterns of agricultural practices throughout Neolithic China. The ultimate purpose is to capture the key changes in the development of early agriculture in China, which were highly likely caused by the climate change. In order to emphasize this point, we have modified the text by saying ‘This is because these broad patterns are not dwelled on any specific site. One can always find multiple sites within one region for cross-checking. Furthermore, data with completely different nature such as stable isotopes of human bones are also combined to justify the archaeological narratives in below’.

In this paper, we choose to avoid complicated statistical analysis for two reasons. One is that we hope to present the data in the most straighforward way, or in other words, let the data speak for itself. Because we use a big-data approach, extra statistical analysis could add more complexities in both presentations and discussions. The current state of analysis is adequate to convey a well-tested argument. Secondly, a significant group of audience to this paper will be archaeologists who normally don’t have strong statistical background. Presenting the data in the most straightforward way can make the argument better conveyed and stimulate more discussion.

2.The available paleoclimate records from the region are much more complicated than what the authors present. Instead, the authors only discuss very broad-scale changes. In some sections they refer to “frequent climate fluctuations”, but what are they? I’m unconvinced, based on the evidence presented, that there are any clear anti-phased precipitation patterns across the region. The authors should critically evaluate all the datasets that they present. I also do not understand the relevance of all the various records presented in Figure 4. For example, how do they interpret percentage of pollen? Any why only present pollen records as precipitation proxies? What about the stable isotope records from the region?

Re: We are sorry for making unnecessary confusion. The term ‘frequent climate fluctuations’ has been replaced by ‘broad climate change’. It is admitted that there is no perfect climate record universally recognized and various problems exist in different proxies. We have incorporated  several new paleoclimate records in the revised manuscript. In fact, either a single point or multiple ones from a whole region suggest that the precipitation gradually increased in the Yellow River basin during the early Holocene, reached the highest value around 8000-5000 BP, and then declined. Although there are relatively fewer studies on the Yangtze River Basin (thus fewer palaeoclimate records) and disagreements in the literature, we found that in the existing studies, the precipitation peaked around 11000-7000 BP, and decreased around 7000-5000 BP but increased again around 5000-4000 BP.

As a specific type of the microfossils preserved in the strata, pollen contains environmental information of different periods, which can reflect the paleoclimate directly or indirectly. The quantitative paleoclimatic reconstruction based on pollen is of great interests for understanding the variation of climate and monsoon evolution during the interglacial period. Whilst it is true to say that the accuracy of quantitative reconstruction of paleoclimate by pollen might be controversial, the significant number of pollen researches focused on the Quaternary has greatly deveoloed the while field, making it pratically possible to reconstruct the Holocene climate in a more accurate way, provided suitable quantitative methods and modern pollen data are applied. We fully take the point raised by the reviewer. The percentage of tree pollen, though closely related to the the vegetation changes in eastern Gansu Province as well as the millet agriculture intensification appears only locally representative. We have therefore removed it in the revised manuscript.

The current pollen-based records (Figure 4d&f) are selected for two reasons. One is that in recent years, the pollen-based method for quantitative palaeoclimatic reconstructions, combined with modern pollen databases, has significantly improved our understanding of past environmental changes (Birks and Gorden, 1985; Herzschuh et al., 2004; Jiang et al., 2006; Xu et al., 2010; Chen et al., 2015). Secondly, the sample resolution of precipitation recorded from the Gonghai Lake and the synthesis of precipitation recorded from the Chaohu Lake, the Gucheng Lake and the Pingwang Lake in the lower Yangtze region is high.

The reason the stalagmite oxygen isotope is excluded from the current study is attributed to the complexity of the variation in the stalagmite isotope in the East Asian monsoon region. More and more scholars argue that the stalagmite oxygen isotope in the East Asian monsoon region is only indicative of the precipitation isotope, rather than precipitation itself (Liu et al., 2015; Chen et al.,2016; Rao et al.,2016).

3.Many sections of the paper need to be reorganized. For example, the temperature discussion starts and stops in various sections. It should be more coherent. There are also numerous typos and grammatical mistakes, which makes the paper very difficult to read in its present form.

Re:Thanks for the comments. We have cleaned up the structures more thoroughly, polished the language and corrected the grammar mistakes as well as typos.

Reference:

Birks, H.J.B., Gordon, A.D., 1985. Numerical methods in Quaternary pollen analysis. Academic Press, London, 317.

Herzschuh, U., Tarasov, P., Wünnemann, B., Hartmann, K., 2004. Holocene vegetation and climate of the Alashan Plateau, NW China, reconstructed from pollen data. Palaeogeogr. Palaeoclimatol. Palaeoecol. 211(1), 1-17.

Jiang, W.Y., Guo, Z.T., Sun, X.J., Wu, H.B., Chu, G.Q., Yuan, B.Y., Hatté, C., Guiot, J., 2006. Reconstruction of climate and vegetation changes of Lake Bayanchagan (Inner Mongolia): Holocene variability of the East Asian monsoon. Quat. Res. 65(3), 411-420.

Xu, Q.H., Xiao, J.L., Li, Y.C., Tian, F., Nakagawa, T., 2010. Pollen-based quantitative reconstruction of Holocene climate changes in the Daihai Lake area, Inner Mongolia, China. J. Clim. 23(11), 2856-2868.

Chen, F.H., Xu, Q.H., Chen, J.H., Birks, H.J.B., Liu, J.B., Zhang, S.R., Jin, L.Y., An, C.B., Telford, R.J., Cao, X.Y., Wang, Z.L., Zhang, X.J., Selvaraj, K., Lu, H.Y., Li, Y.C., Zheng, Z., Wang, H.P., Zhou, A.F., Dong, G.H., Zhang, J.H., Huang, X.Z., Bloemendal, J., Rao, Z.G., 2015. East Asian summer monsoon precipitation variability since the last deglaciation. Sci. Rep. 5, 11186.

Liu,J.B.;Chen,J.H.;Zhang,X.J.;Li,Y.;Rao,Z.G.;Chen,F.H.Holocene East Asian summer monsoon records in northern China and their inconsistency with Chinese stalagmite δ18O records.Elsevier B.V.,2015,148(148).

Rao, Zhiguo , et al. "Investigating the long-term palaeoclimatic controls on the δD and δ18O of precipitation during the Holocene in the Indian and East Asian monsoonal regions." Earth Science Reviews 159(2016):292-305.

Reviewer 2 Report

In this article, Li et al. use 120 or so archaeobotanical records from China to examine how cropping patterns in the millet dominant north and the rice dominant south shift over time as a result of climate change, inferred from paleoclimatic proxies. While their conclusions are not new or innovative as this archaeobotanical work has been done before, the review is useful in that it synthesizes and simplifies the dense archaeobotanical literature to reveal how cropping patterns have shifted at the border of north and south China. However, there are many issues that must first be addressed before this article is suitable for publication, therefore I recommend this article for publication, pending major revisions.

First and foremost, the authors need to be clearer about how they collected their archaeobotanical assemblage and how they are comparing these assemblages across all of China. I find it rather remarkable that, for a paper that uses archaeobotany as its primary dataset, the dataset is only mentioned briefly and as if these data are representative of some inherent truth. These data, like all archaeological data, have flaws, and these flaws should at least be given some due consideration.

Second, the use of the paleoclimatic records is equally problematic. Why were these paleoclimatic records chosen? Is it possible to use these records or others to reconstruct the climate at the boundary zone between north and south China? Many of the paleoclimatic records the authors use here are located far away from the areas of key interest, making their argument weaker.

Third, is it possible to tease out any more interesting conclusions from this dataset? Archaeologists and historians have known for a long time that there is a division between northern and southern China’s dietary history, and that this is related, at least at some degree, to climatic fluctuations. What more can you add to this discussion – I should think it could be more, giving the amount of data that you have available at your disposal?

Before publication, the English language of the article must be significantly improved. There are many grammatical errors, typos, and disjointed sentences and paragraphs. I understand writing in a second language is challenging, but the quality of the writing, at this moment, is unacceptable for publication. Moreover, the figures for this article are rather lackluster. Given that one of the key arguments of this article is the ability to compare how archaeobotanical assemblages change over time as a function of climate, it would be better to be able to view all of the archaeobotanical maps within one figure, as to enable easier comparison. The color scheme between the millet and the rice should be the same in these maps and in the paleoclimate graph.

Finally, there are some typographical errors as well – different font sizes and colors. The authors should endeavor to fix these as well.

Author Response

Dear Editor and Reviewers,

Thanks very much for taking your time to review this manuscript. We thank the reviewers for the time and effort that they have put into reviewing the previous version of the manuscript. Their suggestions have enabled us to improve our work.

Reviewer #2:

1.The authors need to be clearer about how they collected their archaeobotanical assemblage and how they are comparing these assemblages across all of China. I find it rather remarkable that, for a paper that uses archaeobotany as its primary dataset, the dataset is only mentioned briefly and as if these data are representative of some inherent truth. These data, like all archaeological data, have flaws, and these flaws should at least be given some due consideration.

Re: This suggestion is highly important. Thank you very much. We have added a new section to present the potential flaws in the data and discuss why the broad observations are still of scholarly interests.

Obviously, the quality of these data is varied considerably. Franking speaking, the very first level of complexity is actually caused by the archaeological excavations, as some of which are not fully recovered and the samples are only collected from surveyed profiles, thus unable to generate a complete picture of the use of different crops. It is also admitted that data from different sites may derive from rather different approaches and identification of crop remains is often subjected to the analyst’s experience (e.g. wild crops vs domesticated crops). Despite these potential complexities, the broad changes underpinned by our big-data approach are arguably valid and important. This is because these broad patterns are not dwelled on any specific site. One can always find multiple sites within one region for cross-checking with one another. Furthermore, data with completely different nature such as stable isotopes of human bones are also combined to justify the archaeological narratives in below.

2.The use of the paleoclimatic records is equally problematic. Why were these paleoclimatic records chosen? Is it possible to use these records or others to reconstruct the climate at the boundary zone between north and south China? Many of the paleoclimatic records the authors use here are located far away from the areas of key interest, making their argument weaker.

Re:Thanks for the suggestion. The very first principal of choosing these palaeoclimate records is based on the chronology and regionality. In terms of chronology, they have to be dedicated to the periods of our research interests on one hand and contain adequate chronological resolution on the other hand. It should be stressed that in the revised version we have included new and multiple records for cross-checking with one another. The conclusions are therefore further justified.

It is great to see that the reviewer points out the issue of regional representation. We have modified our figures by adding the geographical locations of the climate records on the map. One can easily see that they are all located in the areas in our discussion and can be considered representative. Thanks to the reviewer’s comment, the new figure not only addresses the issue but also strengthens the argument.

We are also very interested in the boundary zone. However, no high-quality climate records specific to this region allows us to carry out in-depth research on the relationship between agriculture and climate. We have to await for more records to be published in the future.

3.Is it possible to tease out any more interesting conclusions from this dataset? Archaeologists and historians have known for a long time that there is a division between northern and southern China’s dietary history, and that this is related, at least at some degree, to climatic fluctuations. What more can you add to this discussion – I should think it could be more, giving the amount of data that you have available at your disposal?

Re:Thanks for the interesting suggestion. We have sharpened our conclusion to highlight the fact that the long-term argument made by historians is merely one static point in the overall development of Chinese agriculture. It is a highly dynamic process over the majority of Holocene and climate change played an essential role.

4.Before publication, the English language of the article must be significantly improved. There are many grammatical errors, typos, and disjointed sentences and paragraphs. I understand writing in a second language is challenging, but the quality of the writing, at this moment, is unacceptable for publication. Moreover, the figures for this article are rather lackluster. Given that one of the key arguments of this article is the ability to compare how archaeobotanical assemblages change over time as a function of climate, it would be better to be able to view all of the archaeobotanical maps within one figure, as to enable easier comparison. The color scheme between the millet and the rice should be the same in these maps and in the paleoclimate graph.Finally, there are some typographical errors as well – different font sizes and colors. The authors should endeavor to fix these as well.

Re:Thanks for the comments. We have polished the language and part of statements. We also have tried to draw a new figure with all of the archaeobotanical maps as suggested which is shown below, but we thought it's too confusing, so we decided to show the archaeobotanical evidence in different periods. We have changed the color of millet and rice in fig.4, thanks a lot for the reminder.
